# Bridging between Mouse and Human Enhancer-Promoter Long-Range Interactions in Neural Stem Cells, to Understand Enhancer Function in Neurodevelopmental Disease

**DOI:** 10.3390/ijms23147964

**Published:** 2022-07-19

**Authors:** Romina D’Aurizio, Orazio Catona, Mattia Pitasi, Yang Eric Li, Bing Ren, Silvia Kirsten Nicolis

**Affiliations:** 1Institute of Informatics and Telematics (IIT), National Research Council (CNR), 56124 Pisa, Italy; orazio.catona@gmail.com; 2Dipartimento di Biotecnologie e Bioscienze, University of Milano-Bicocca, 20126 Milano, Italy; m.pitasi@campus.unimib.it (M.P.); silvia.nicolis@unimib.it (S.K.N.); 3University of California San Diego, La Jolla, CA 92093, USA; yal054@health.ucsd.edu (Y.E.L.); biren@health.ucsd.edu (B.R.)

**Keywords:** enhancers, non-coding DNA, genome-wide association studies (GWAS), copy number variants (CNV), long-range interactions, neural stem-cells, neurodevelopmental diseases, single-nucleotide polymorphisms (SNP)

## Abstract

Non-coding variation in complex human disease has been well established by genome-wide association studies, and is thought to involve regulatory elements, such as enhancers, whose variation affects the expression of the gene responsible for the disease. The regulatory elements often lie far from the gene they regulate, or within introns of genes differing from the regulated gene, making it difficult to identify the gene whose function is affected by a given enhancer variation. Enhancers are connected to their target gene promoters via long-range physical interactions (loops). In our study, we re-mapped, onto the human genome, more than 10,000 enhancers connected to promoters via long-range interactions, that we had previously identified in mouse brain-derived neural stem cells by RNApolII-ChIA-PET analysis, coupled to ChIP-seq mapping of DNA/chromatin regions carrying epigenetic enhancer marks. These interactions are thought to be functionally relevant. We discovered, in the human genome, thousands of DNA regions syntenic with the interacting mouse DNA regions (enhancers and connected promoters). We further annotated these human regions regarding their overlap with sequence variants (single nucleotide polymorphisms, SNPs; copy number variants, CNVs), that were previously associated with neurodevelopmental disease in humans. We document various cases in which the genetic variant, associated in humans to neurodevelopmental disease, affects an enhancer involved in long-range interactions: SNPs, previously identified by genome-wide association studies to be associated with schizophrenia, bipolar disorder, and intelligence, are located within our human syntenic enhancers, and alter transcription factor recognition sites. Similarly, CNVs associated to autism spectrum disease and other neurodevelopmental disorders overlap with our human syntenic enhancers. Some of these enhancers are connected (in mice) to homologs of genes already associated to the human disease, strengthening the hypothesis that the gene is indeed involved in the disease. Other enhancers are connected to genes not previously associated with the disease, pointing to their possible pathogenetic involvement. Our observations provide a resource for further exploration of neural disease, in parallel with the now widespread genome-wide identification of DNA variants in patients with neural disease.

## 1. Introduction

The study of the genetic basis of human inherited disease has so far centered mostly on the study of protein-coding genes, with the identification of disease-causing mutations involving gene sequences (exons), or sequences involved in the splicing of primary transcripts into mature mRNA. The potential of non-coding DNA (representing 98% of our genome) to contribute to genetic disease has remained comparatively unexplored. Further, it was estimated that only 42% of patients with severe developmental disorders carry pathogenic de novo mutations within coding sequences [1], further encouraging consideration of the non-coding genome as an important territory to chart for mutations. Genome-wide association studies (GWAS) identified thousands of sequence variants located in non-coding regions, possibly corresponding to gene regulatory elements, which might be contributing to disease by affecting the regulated gene [2,3]. The non-coding genome is known to contain transcriptional regulatory DNA elements, enhancers, and promoters, fundamental to orchestrate regulated gene expression during development and differentiation. Some “historical” examples demonstrate well the ability of enhancers and promoters to contribute to genetic disease (e.g., the beta-globin genes *LCR* and gamma-globin regulatory regions for hemoglobinopathies; enhancers of the *Sox9* and *SHH* genes for sex-determination and mesodermal pathologies [4]. However, difficulties in identifying functional enhancers within the non-coding genome have until recently represented a hurdle in searching for their role in disease pathogenesis in a more generalized way [5]. The advent of next-generation sequencing (NGS)-based functional genomics represented a major leap forward in the identification of enhancers, and in their annotation to the gene(s) they regulate. ChIPseq, and, more recently, CUT&RUN, allow for the mapping of epigenetic enhancer marks, such as histone modifications (H3K27Ac, H3K4me1), together with transcription factor (TF)-binding sites, in chromatin at the genome-wide level (https://www.encodeproject.org/, accessed on 15 January 2022) [6,7,8]. Chromatin Interaction Analysis by Paired-End Tagging (ChIA-PET), and other related methods, identify long-range interactions between gene promoters and distant DNA sequences, carrying epigenetic enhancer marks, together with bound TF. In particular, RNApolIIChIA-PET identifies long-range interactions in chromatin, involving RNApolII, the enzyme that transcribes genes. Remarkably, RNApolII ChIA-PET showed that, at the genome-wide level, a given enhancer is very often not connected to the nearest promoter, but rather to a more distant one(s), skipping genes in between [9]. These observations point to the need for experimental determination of long-range interaction maps, to correctly annotate a given enhancer to the promoter it interacts with (often far away).

Recently, we performed RNApolIIChIA-PET analysis of mouse brain-derived neural stem cells (NSC) cultured from the mouse perinatal (postnatal day 0, P0) brain, coupled to ChIPseq profiling of epigenetic enhancer marks (H3K27Ac; H3K4me1), binding sites for the *SOX2* and *FOS* transcription factor, and RNAseq analysis of gene expression in normal versus *Sox2*-deleted NSC [7,10]. This identified thousands of interactions connecting gene promoters to distal enhancers, carrying epigenetic enhancer marks (and TF binding sites). Remarkably, 14 out of 15 enhancers tested in transgenesis assays directed gene expression to the developing brain; further, genes downregulated following *Sox2* deletion were highly enriched in long-range interactions (LRI) with enhancers carrying epigenetic enhancer marks, and *SOX2* binding sites. These observations pointed to the functional relevance of the identified enhancer–promoter interactions for gene regulation [10].

In the present paper, we discover, in the human genome, thousands of DNA regions syntenic with the interacting regions (enhancers and connected promoters) previously identified, in mice, by RNApolII-ChIA-PET [10]. We term these regions human–mouse syntenic long-range interaction regions (hmsLRI). Within hmsLRI, we define regions homologous to mouse regions carrying epigenetic enhancer marks (termed hmsLRI-Enhancers, hmsLRI-E). We find that hmsLRI-E are highly enriched (within their mouse counterparts) in candidate cis-regulatory elements (cCRE), previously defined on the basis of their chromatin accessibility, in single-nuclei analyses, in a variety of adult brain cell types [11]; in agreement with this, most (95.5%) hmsLRI-E also carry epigenetic enhancer marks in human neural cells.

We further relate these regions to sequence variants (single nucleotide polymorphisms, SNP; copy number variations, CNV), that were previously associated with neurodevelopmental disease (NDD) in humans. We find that these sequence variants associated with NDD are highly represented within hmsLRI. We document various cases in which the genetic variant, associated in humans to NDD, affects an enhancer, or a promoter, involved in LRI. These observations provide novel hypotheses regarding the pathogenesis of the NDD associated with those variants. Further, we propose that hmsLRI may provide new hypotheses as to the possible location of new pathogenic sequence variants in NDD, to be experimentally tested, in the future (e.g., by targeted sequencing of enhancers connected to genes whose mutation, in exons, are known to cause NDD).

## 2. Results

### 2.1. Identification of Human DNA Regions Syntenic with Mouse Enhancer–Promoter Long-Range Interactions, and of Putative Human Enhancer Sequences Therein

We sought to identify, within the human genome, DNA regions syntenic with the long-range interactions (LRI) previously identified by RNApolII-ChIA-PET in mouse NSC, connecting gene promoters with distal “epigenetic enhancers”, i.e., regions marked by H3K4me1 and H3K27Ac [12,13] (Methods) (Figure 1A). We first identified, within mouse LRI, the “non-promoter” interaction anchors [10] carrying epigenetic enhancer marks. “Anchors” are defined as DNA regions, identified by RNApolII-ChIA-PET to interact with distant DNA regions; “Promoter anchors” contain a gene promoter, “non-promoter” anchors do not [10]. We distinguished between regions carrying both H3K27Ac and H3K4me1 (“active enhancers”) and regions carrying H3K4me1 only (“poised enhancers”, representing enhancers that were proposed to become fully active following cell differentiation) [12,13] (Figure 1B; Methods). We reciprocally mapped mouse enhancers and promoters of each LRI to their syntenic human regions by the Liftover tool (see Methods); we termed these human regions human–mouse syntenic long-range interaction regions (“hmsLRI”). We further distinguished within the hmsLRI the sequence homologous to the mouse epigenetic enhancer sequence (i.e., carrying active or poised enhancer marks); we term these sequences Enhancers within hmsLRI (hmsLRI-E) and hmsLRI-E-a (active) or hmsLRI-E-p (poised) according to the epigenetic marks (Figure 1B; Table 1). Promoters, as they are more conserved, showed a higher percentage of continuous overlap than enhancers, as the latter regions most often encompass intergenic and intragenic regions so they are less likely to be completely conserved in humans (Appendix A). Overall, out of 10,994 mouse LRI originally identified in [10] (7554 involving active enhancers, and 3440 involving poised enhancers), 7698 LRI could be re-mapped onto the human genome regarding both the promoter and the enhancer (Figure 1B, “P+E”), with more than 50% of bases “lifted over” to the human genome; these represent about 70% (72.7% for active and 64.1% for poised, respectively) of the LRI originally identified in mouse (Table 1 and Appendix A, Figure 1C). These human regions (promoter+enhancer, P+E), corresponding to the mouse LRI via Liftover, are the hmsLRI (Table 1, Figure 1B). Interestingly, for 33 mouse LRI (26 involving active and 7 involving poised enhancers), both enhancer and promoter could be re-mapped, but, in humans, the corresponding sequences were located on two distinct chromosomes (interchromosomal LRI in Table 1, Figure 1B, “split P and E”) implying a probable disruption of the mouse 3D chromatin interaction.

We also identified 493 enhancers (358 active, 135 poised), involved in LRI in mice, which “lifted over” to the human genome, but for which the corresponding promoter anchor in humans could not be identified as part of a syntenic region (Table 1, Figure 1B, “E only”). On the other hand, for a total of 2308 mouse LRI, only the promoter sequences were conserved in a syntenic region of the human genome (Table 1, Figure 1B, “P only”). These cases (E only, P only) may result from loss, in evolution, of the promoter or enhancer sequences present in the mouse (and presumably in a common ancestor), or from the gain of enhancers or promoters in mice, but not in humans; they may, however, also result from the splitting, in humans, of the single enhancer/promoter anchor sequence present in mice, with the two split parts ending up on different chromosomes (or on distant parts of the same chromosome), thereby failing to reach the 50% overlap required to qualify as re-mapped by the chosen Liftover parameters. In the present work, we shall focus our attention on those 7698 LRI for which the mouse enhancer and interacting promoter could both be re-mapped in the human genome, and were part of a syntenic region, hence they were located on the same chromosome in both mouse and human (listed in Appendix A). As expected, the number of LRI conserving the promoter regions exceeded those for which only the enhancer was conserved between the two species.

The hmsLRI-E showed significantly higher levels of sequence conservation among 100 vertebrate genomes as compared to random shuffled genomic regions (Figure 2A; Methods). Each of the identified hmsLRI regions were annotated by comparison with the gene coordinates. To annotate hmsLRI-P, priority was given to check the distance from any transcription start site (TSS) within a window of 5 kb centered in the TSS for every gene (Methods). Most of hmsLRI-E were intronic or intergenic (Figure 2B). A total of 4635 hmsLRI-E were intragenic (i.e., intronic or exonic) and are reported in Appendix A. Of note, around 50% (3797) of the syntenic enhancer–promoter interactions spanned multiple genes or the enhancers were in the intron of another gene. The full list of the identified hmsLRI and associated gene annotation is reported in Appendix A, and a summary is reported in Table 1. We also created, for the WashU human genome browser, a track depicting the location of the identified hmsLRI, and hmsLRI-E, with hmsLRI represented by loops.

### 2.2. HmsLRI and hmsLRI-E Overlap with DNA Regions Previously Shown to Be Involved in Long-Range Interactions, and to Carry Epigenetic Enhancer Marks, in Human Neural Cells

Before further analyzing hmsLRI, we attempted to validate them by comparison with previously mapped interactions in human neural tissues from different brain regions and developmental stages. First, we evaluated if hmsLRI were located within topologically associating domains (TADs), or spanned a TAD boundary (intra- or inter-TAD, respectively), as identified by the PsychENCODE Consortium [14] and Won et al. [15] (Methods). Of note, for 7232 (93% of total 7698) unique hmsLRI, both the promoter and enhancer were found to be located within the same TAD in at least one cell type we analysed (89% were in human adult dorsolateral prefrontal cortex (PFC) of PsychENCODE, 87% and 86% in fetal brain cortical plates (CP) and germinal zones (GZ), respectively, [15]. A small fraction (2.7% total unique; 3.8% for PFC, 9.3% for CP and 11.7% for GZ) of the total set were inter-TAD while 0.3% resided out of the considered TADs.

Next, we investigated whether the hmsLRI-E also carry epigenetic enhancer marks in human neural cells. To this end, we used the H3K4me1 and H3K27Ac marks obtained from ChIP-seq experiments by the Roadmap Epigenome Project [16] on a plethora of different brain tissues and developmental stages, or cultured neural stem/progenitor cells from different origins. Between 79.6% and 94.4% of hmsLRI-E carried epigenetic H3K4me1 marks, depending on the tissue/cell type analyzed (see Figure 3A); cumulatively, 99.5% of hmsLRI-E showed epigenetic enhancer marks in at least one neural tissue analyzed (Figure 3B). The significance of such an outcome was proved by comparison with the % of matches with random shuffled genomic regions which were simulated with similar length and chromosomal distribution of our hmsLRI. The H3K4me1 peaks corresponding to potential enhancers were available for each neural tissue while H3K27ac only for a subset for which we could distinguish between active and poised enhancers. For these cases, most of the hmsLRI-E were active also in humans since they encompassed regions with both histone modifications (see Figure 3A).

We further investigated the concordance between hmsLRI-E and epigenetic enhancer marks in each tissue at base level by taking into account the width of the overlaps and found that, with the exception of H1 and H9 derived NPC, they share more than 1000 bp (Figure 3C). Lastly, we calculated the proportion of hmsLRI-E bps covered by the epigenetic marks out of the total length of the matched hmsLRI-E and compared it to those from the simulated shuffled set (Figure 3D). On average, slightly less than half of hmsLRI-E width was marked (min 0.36, max 0.49) for all neural tissues, the most concordant being the dorsolateral prefrontal cortex and anterior caudate. This finding importantly suggests that a critical functional feature of enhancer activity is conserved between mouse and human enhancer sequences.

Further, we evaluated whether enhancer–promoter connections observed in mouse neural cells are also maintained within their hmsLRI counterparts. For this, we compared hmsLRI with five previously published human datasets of genome-wide long-range interactions, obtained mainly using HiC, and H3K4me3-mediated chromatin interactions to histone-marked active promoter regions, by PLAC-seq assays (Table 2). We found that, cumulatively, up to 51% of hmsLRI were represented within human experimentally detected long-range interactions. Notably, the number of interactions varied a lot between the different datasets, and a large proportion of hmsLRI identified in a given dataset were not identified in one or more of the other datasets, implying that the published HiC/PLAQ-seq data do not exhaustively represent the totality of interactions. This might reflect technical differences between different HiC experiments, as well as the fact that the source of the neural tissue/cells is quite diverse in the reported datasets, and some interactions may therefore be present in only a subset of the analyses reported. Other possible explanations are presented in the Discussion section. We conclude from the analysis in Table 2 that the cumulative representation of hmsLRI (51%) within published interactions datasets is likely a minimum estimate, that may increase when additional tissues/cell types are analyzed in more detail. Taking this result together with the extremely high conservation of epigenetic enhancer characteristics of hmsLRI-E in human neural cells, we are now confident that our present data may be useful for further studies.

### 2.3. Further NDD-Related Functional Annotation of hmsLRI-E

To further complement the identification of hmsLRI-E with useful annotations and supporting evidence about the potential functional role of those regions specifically in the NDD context, we also checked hmsLRI-E overlaps with VISTA enhancers. The VISTA enhancer browser (https://enhancer.lbl.gov/, accessed on 25 October 2021) [19] collects experimentally validated non-coding fragments with gene enhancer activity in humans and mice; in our previous work, we had found that VISTA enhancers are significantly represented within ChIA-PET-identified mouse enhancers [10]. Here, we also looked for evidence of confirmed gene enhancer activities for hmsLRI-E regions. We found a total of 212 VISTA positive enhancers among the hmsLRI-E, of those 182 were active enhancers and 30 were poised enhancers (see Appendix A).

Furthermore, we exploited the Gene2Phenotype databases (http://www.ebi.ac.uk/gene2phenotype, accessed on 12 November 2021) to check for the presence of high-confidence neurodevelopmental (brain) and eye disorder genes (DD panel and Eye panel in the dataset, respectively) among those involved in hmsLRI. We also annotated our hmsLRI for their overlap with the DDG2P dataset (https://www.deciphergenomics.org/ddd/ddgenes, accessed on 12 November 2021). DDG2P includes genes with curated annotation of disease association level of certainty that the gene causes a developmental disease (e.g., definitive, strong, moderate...), known consequences of pathogenic mutations in the gene, as well as information about the tissues affected by gene mutations (Methods). In Appendix A, we annotated each human gene involved in hmsLRI, connected with the enhancer or the promoter anchor, if it was reported in the DDG2P database and specified the level of disease association. Regarding the Gene2Phenotype database, a total of 321 in over 1474 genes involved in brain, cognition and eye disease, and 112 in over 667 genes involved specifically in eye disease, were found involved in hmsLRI (see Appendix A).

### 2.4. Mouse Enhancers and hmsLRI-E Are Enriched in Brain Candidate Cis-Regulatory Elements (cCRE)

In a recent study, Li et al. [11] identified a set of candidate cis-regulatory elements (cCRE) in the mouse genome, by combining single-nucleus assays for transposase-accessible chromatin (sn-ATAC-seq) and single-cell RNA sequencing (scRNA-seq), from more than 800,000 cells from 45 different regions of the adult mouse brain. They identified 491,818 cCRE within 160 distinct cell types. We compared our mouse enhancers, involved in LRI, to mouse cCRE. We found that more than 95% of our mouse enhancers (10,501 in over 10,994) overlap with cCRE (97.6% of the active enhancers, 7370 in over 7554, and 91% of the poised ones, 3131 in over 3440, respectively). Overall, they spanned all the 160 cell types with different levels of occurrence ranging from 12% to 78%. Accordingly, all 43 subclasses were present with variable fractions of enhancers overlapping with cCRE (12% to 76%, Figure 4). Remarkably, the most enriched cell types encompass Radial Glia-like Stem Cells (RGL, corresponding to the identity of our forebrain-derived cultured NSC), and other proliferating neural precursors (such as oligodendrocyte progenitor cells, OPC), but also many differentiated cell types within the in vivo brain (Figure 4). Interestingly, the most represented cell types include various types of GABAergic neuroblasts and neurons, as well as astrocytes (ASC) and oligodendrocytes (OGC) (Figure 4), which correspond to the cell types that our NSC generate upon in vitro differentiation [7,8,10,20]. Glutamatergic neurons, though present, are less represented (Figure 4).

For 7451 in over 10,501 mouse enhancers overlapping a cCRE, we could also associate by synteny an hmsLRI-E. In particular, we observed that almost all of our hmsLRI-E (97%, 7451 in over 7698) overlayed with one or more cCREs, providing additional evidence of their potential regulative role also in humans. We annotated the hmsLRI-E within Appendix A by reporting the original cCRE ID (as in Appendix A of the Li et al. [11] paper) of the cCREs overlapping the human LRI-E identified by synteny with the mouse enhancers involved in LRI.

### 2.5. HmsLRI-E Regions Overlap with Non-Coding Risk Variants Associated with NDD

We next sought to relate the identified hmsLRI-E to sequence variants previously found in humans by genome-wide association studies (GWAS) to be associated with neurological diseases and traits. Most such variants are located in the non-coding part of the genome, often lacking functional annotation, making it difficult to interpret their meaning [5]. To test whether our interaction maps could help the interpretation of some of these variants, we checked their presence within our hmsLRI-E to provide a way to functionally connect these variants to a specific gene promoter, via the long-range interaction. We thus searched our hmsLRI-E for the GWAS-derived variants with posterior probabilities of association (PPA) score larger than 1% with schizophrenia (SCZ), bipolar disorder (BP), and intelligence (I). These potential causal variants were derived from the previous analyses by [11]. In all three datasets, we identified variants overlapping hmsLRI-E (Table 3 and Appendix A for details).

Significantly, 26 risk variants for schizophrenia were located in 31 hmsLRI-E regions and most of them were connected with different genes previously proposed to contribute to the pathology (*TRIM8*, *SREBF2*, *LINC00461*, *ETF1*, *CTNND1*, *PITPNM2*, *TCF4*, *FOXP1*, *NSD3*, Table 3 and Appendix A for details). This suggests that the variant itself, or a closely associated mutation, may contribute to the disease by affecting the transcription of the connected gene. In particular, a cluster of five different risk variants was within five different hmsLRI-Es (three of them were intergenic and two overlaid 2222he 3′ UTR of the ARL3 gene) which were connected to the promoter of the *TRIM8* gene (Figure 5). They may all be involved in the transcriptional deregulation of this gene, a proposed contributing factor to schizophrenia [21]. Interestingly, other variants were found in hmsLRI-E connected to genes that had not been previously associated with schizophrenia (*KIZ*, *CDKAP2*, *EPN2*, *MAPK7*, *B9D1*, Table 3). It is possible that these variants (or closely associated sequence variants) contribute to the disease by deregulation of these connected genes, pointing to their possible contribution to the pathology. Of note, among the hmsLRI-E that contain schizophrenia-associated risk variants (Table 3), 17 also overlapped with the human counterpart of cCREs (Table 3 and Appendix A); this indicates that, in mice, the enhancer involved in the LRI is in an open configuration in adult brain cell types. Interestingly, these enhancers were associated with six different genes by our LRI. In the Bipolar disorder dataset, four risk variants were found to overlap with hmsLRI-E which were connected to the *NR2C2AP* gene (two variants), the *MPP2* gene (one variant), and the *LMS12* gene (two variants) (Table 3). In the Intelligence dataset (collecting variants that segregate with different levels of intelligence), 38 risk variants were into 45 enhancer regions connected to 25 different genes (Appendix A), interestingly including *FOXP1* (https://www.deciphergenomics.org/gene/FOXP1/overview/clinical-info, accessed on 15 December 2021) (already involved in language impairment and autism), *JARID2* (https://www.deciphergenomics.org/gene/JARID2/overview/clinical-info, accessed on 15 December 2021), *PTCH1* (https://www.deciphergenomics.org/gene/PTCH1/overview/clinical-info, accessed on 15 December 2021), and others (Table 3). Overall, our data suggest that the identified variants are associated to enhancers, that may regulate the gene to which they are connected via LRI. Whether the sequence variant itself causes deregulation of the connected gene, or whether this is caused by a different mutation in close linkage with it, remains to be determined. To begin to investigate the possible pathogenic meaning of some of the variants overlapping our enhancers, we tested whether the variants overlapping the *TRIM8*-connected enhancers affect potential transcription factor binding sites, as reported in the ENCODE database of TF DNA-binding motifs. In Figure 5B, we show that sequence variants overlapping with three different hmsLRI-E alter the recognition sequence for TF on DNA; in two cases the mutation is predicted to lead to loss or reduction of TF binding, in one case to lead to the creation of a TF binding site. These findings provide new, testable hypotheses regarding the functional effect of GWAS-derived sequence variants overlapping our enhancers.

**Figure 5 ijms-23-07964-f005:**
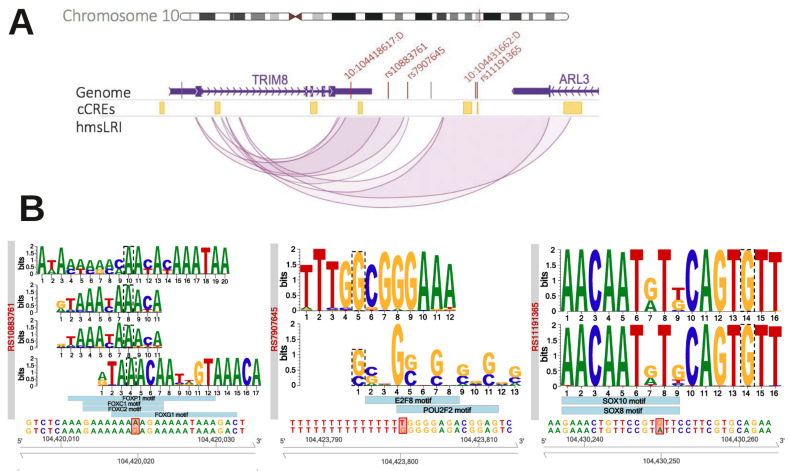
**(A) GWAS-identified risk variants associated with schizophrenia overlap with five different hmsLRI-E connected with the*****TRIM8*****gene.** Visualization of the human *TRIM8*-*ARL3* locus, located on 10q24.32. In the middle track, in yellow, cCREs identified by (Li et al., 2021) [11] lifted over to the human genome. In the lower track, our hmsLRI: enhancers are represented in their full length, while promoter as their middle point. In the top genome track, vertical bars represent SNPs associated to schizophrenia according to (Li et al., 2021) [11]; the ones shown in red fall into one of our hmsLRI-E connected to the *TRIM8* promoter. **(B) Neural TF binding site alteration by the risk variants.** Alteration by the risk variants of putative neural TF binding sites, as predicted by motifbreakR (Coetzee et al., 2015) [22]. From below upwards: reference DNA sequence and SNP alteration, highlighted in red; light blue boxes represent motifs of transcription factors whose binding is predicted to be strongly influenced by the SNP; detail of the binding motif, retrieved from ENCODE. The motif nucleotide affected by the SNP is surrounded by a dotted line; if the nucleotide is the same as the reference genome, the binding site is predicted to be lost, otherwise it is gained: rs10883761 and rs1191365 disrupt potential binding sites of the TF represented, while rs7907645 leads to the acquisition of the indicated potential novel TF binding sites.

**Table 3 ijms-23-07964-t003:** **Sequence variants associated with Schizophrenia and Bipolar disorder overlap with human putative enhancers, involved in human–mouse syntenic long-range interactions (hmsLRI-E, see Figure 1, Table 1)**. “Gene” (first column) is the gene whose promoter is involved in the human–mouse syntenic long-range interaction (hmsLRI); the sequence variants are located within the connected hmsLRI-E. We searched our hmsLRI-E for the GWAS-derived variants with posterior probabilities of association (PPA) score larger than 1% with schizophrenia and bipolar disorder; these potential causal variants were derived from the previous analyses by (Li et al., 2021) [11] (see their Appendix A). * PsychENCODE association of genes to schizophrenia was by HiC connection between the variant-containing DNA region and the gene promoter, and by eQTL data (see Wang et al., 2018 [14] and (http://resource.psychencode.org/, accessed on 4 March 2022).

Gene	Risk Variants Overlapping hmsLRI-E	Number hmsLRI Involved	Gene Previously Associated with the Disease	Associated to SCZ in PsyschENCODE Dataset *	Gene Product	Reference
SCHIZOPHRENIA						
TRIM8	5	5	Yes (by association studies)	No	Tripartite Motif Protein (E3 Ubiquitin-protein ligase?)	[21]
KIZ	1	1	Yes (by association with SNP)	No	Centrosomal protein	[23] [24]
SREBF2	1	1	Yes (by association with intronic variant SNP)	No	Transcription factor	[25]
LINC00461	1	1	Yes (by association studies)	No	Long non-coding RNA	[26]
ETF1	2	1	Yes (by association studies)	No	Translation termination factor	[27] [28]
CTNND1	1	1	Yes (by association studies)	No	Catenin (cell adhesion, signal trasduction)	[29]
PITPNM2-AS1	1	1	Yes (PITPNM2) (by association studies)	Yes	Antisense RNA	[29]
CDK2AP1	2	1	Yes (by association studies)	Yes	Cyclin-dep kinase-assoc protein	[30]
EPN2	2	3	No	Yes	Eprin (clathrin interactor)	
MAPK7	1 (shared with EPN2)	1	No	No	Mitogen-activated protein kinase	
B9D1	1 (shared with EPN2)	1	No	No	B9-domain-cont protein (ciliogenesis)	
TCF4	4	4	Yes (by association with intronic variant SNP)	No	Transcription factor	[31]
FOXP1	1	1	Yes (by association studies)	Yes	Transcription factor	[32]
LETM2	2	1	No	Yes	Transmembrane protein	
NSD3	2	1	Yes (by association studies)	Yes	Histone H3 methyltransferase, nuclear receptor binding protein	[33]
H2AC20	1	1	No	No	Replication-dependent histone	
BIPOLAR DISORDER						
NR2C2AP	2	1	No		Nuclear receptor associated protein	
MPP2	1	1	No		Membrane protein; synaptic function	
LSM12	2	1	No		Receptor channel	

### 2.6. CNVs Associated with NDD Involve hmsLRI and Can Disrupt Enhancer–Promoter Interactions

We then considered clinically relevant Copy Number Variants (CNV) identified in NDD cohorts with the aim of drawing a possible causal connection with the loss/gain of a copy of a DNA region involving hmsLRI. Indeed, there are reports of neural disease caused by genomic rearrangements affecting the regulation of specific genes, without altering their coding regions; it is possible that these rearrangements disrupt gene connections to their enhancers, or the enhancers themselves (see [34,35,36,37,38] in relation to *FOXG1*, *Bcl11a*, *Arid1A*, *AKT3*).

We analyzed two datasets of CNVs associated with NDD [39,40] for possible overlaps with our hmsLRI, and in particular with hmsLRI-E. The genome-wide catalogue of rare CNV generated by Zarrei et al. [40] includes the genotypes of 2691 subjects affected by autism spectrum disorder (1838 ASD), attention deficit hyperactivity disorder (424 ADHD), schizophrenia (204 SCZ), and obsessive-compulsive disorder (222 OCD) encompassing 55,256 copy number variants. Similarly, the dataset by Coe et al. [39] includes CNVs associated with neurocognitive disorders, developmental delay and/or ASD for a total of 29,085 primarily pediatric cases in comparison with 19,584 adult population controls. We evaluated the representation of our hmsLRI within the CNVs harbored by affected cases in both datasets (Zarrei, Coe) and found a statistically significant enrichment (p<0.0001, according to the hypergeometric test) for both, when compared to the control cohort in Coe’s study, in agreement with hmsLRI containing genes relevant for the pathology, together with their relevant enhancers.

The full list of hmsLRI showing altered copy number (loss or gain) in Zarrei and Coe’s datasets is reported in Appendix A. AnnotSV has been used to annotate the pathogenicity of every CNV (Appendix A).

A particularly interesting class of CNVs is that in which the involved DNA region encompasses the enhancer (hmsLRI-E), but not the promoter connected to it (Appendix A). Thus, we focused our attention on those alterations which were also present exclusively among the affected subjects compared with the controls. An example of such CNV (Figure 6A) consists of deletions involving a non-coding DNA region 3′ to the *FOXG1* gene, but not the *FOXG1* gene itself. These deletions were found in two different patients, affected by ASD forms with characteristics in common to the disease caused by mutations (CNV) affecting the *FOXG1* gene itself. Interestingly, we find that two such deletions overlap an hmsLRI-E, showing evolutionary conservation through multiple species as reflected by a high PhastCons score; further, the corresponding mouse enhancer overlaps cCREs (see above) active in various adult brain cell types (Figure 6A). We note that other hmsLRI are identified for *FOXG1*, together with their associated hmsLRI-E (Figure 6A), defining further candidates to investigate for the presence of pathogenic mutations in ASD in the future.

Further interesting cases include those spanning over hmsLRI-E connected to the promoter of the *TRIO* and *PHF21A* genes; both of these genes are linked to NDD according to the DDG2P database. In particular, one hmsLRI-E (active wTR2_9924 chr11:46315791– 46318951, Appendix A), connected via long-range interaction to *PHF21A*, overlaps two 50 kb deletions characterized in SCZ patients reported in the Zarrei database (Figure 6B, Appendix A). *PHF21A* is part of a transcriptional complex regulating neuronal gene expression and its mutations or deletions have been associated with NDD [41]; intriguingly, the two deletions encompass part of *CREB3L1*, a gene which is, itself, not linked to NDD, and not highly expressed in neural cells according to the Human Protein Atlas Database [42]. These observations suggest that the observed deletions within the region including *CREB3L1* might contribute to the NDD phenotype mainly by causing the loss of the *PHF21A*-connected enhancer, hence the transcriptional deregulation of *PHF21A*.

Regarding the *TRIO* gene, the two connected hmsLRI-E (Appendix A) are intergenic and altered by distinct CNVs reported in the Zarrei dataset. Specifically, a 23 kb deletion encompassing the first enhancer region (active wTR1_2401 chr5:14118146–14120275 and active wTR2_4867 chr5:14118458–14119337, Appendix A), and not any coding region, is present in one SCZ patient; the second *TRIO* enhancer (poised wTR3_2113 chr5:13976963–13977535, Appendix A), altered in two ASD siblings carrying a duplication (~130 kb), is located in an intergenic region (Appendix A). While the deletion might act by removing an enhancer, thereby causing transcriptional downregulation of the gene connected (in normal circumstances) to the enhancer, it is possible that the duplication might act (at least in part) by increasing enhancer dosage; the second enhancer copy might also interact (together with the first one) with the gene promoter, thereby causing increased gene expression. Future work will allow the experimental testing of these hypotheses. The *TRIO* gene encodes a GEF (Guanine Exchange Factor) that promotes axon guidance and neurite outgrowth [43]; *Rho* and *Rac* being its downstream effector, it plays a key role in p75^NTR^ complex, regulating and coordinating actin remodelling by its GEF activity in hippocampal neurons, within which it is localized in actin-rich protrusions [44]. A recent study including individuals with pathogenic variants of the *TRIO* gene highlighted a good genotype–phenotype correlation: missense mutations in the *RAC1*-activating GEF domain cause mild intellectual disability and microcephaly, missense mutations in the seventh spectrin repeat at the N-terminus underlie a phenotype characterized by severe intellectual disability and macrocephaly, while non-sense mutations in the *TRIO* sequence give rise to a more variable phenotype; interestingly, 31% of patients with pathogenic variants of the *TRIO* gene showed autistic traits [45]. A further study suggests that rare variants in the neurotrophin signaling pathway, in which p75 and *TRIO* are key components, are implicated in schizophrenia risk [45,46]. These pathways are important in cell migration, a key neurodevelopmental process altered in many NDD, including ASD and SCZ [46]; as the disruption of the enhancer–promoter connectivity caused by CNVs presumably leads to a transcriptional deregulation of *TRIO*, the entire canonical cell migration pathway might be affected, hence providing a link between the CNVs and the NDD phenotype observed.

A further example of interesting hmsLRI-E overlapping microdeletions is represented by two intronic enhancers of *ZEB2* (i.e., hmsLRI-E wTR1_3899, including wTR1_3900, Figure 6C), a gene whose mutations cause Mowat–Wilson Syndrome, characterized by intellectual disability and microcephaly [47] and recently implicated in human-specific neurodevelopmental processes [48]. An intronic microdeletion (14 kb) overlapping these hmsLRI-E (chr2:145186591–145201317) is reported in the DECIPHER database, in an individual with intellectual disability (DECIPHER #360809). The deleted hmsLRI-Es overlaps a VISTA enhancer, active in the developing CNS (VISTA hs407, https://enhancer.lbl.gov/cgi-bin/imagedb3.pl?form=presentation&show=1&experiment_id=407&organism_id=1, accessed on 25 October 2021). The deletion, via CRISPR, of 961 bp encompassing about two-thirds of hmsLRI-E wTR1_3900 (Figure 6D), was reported to significantly reduce *ZEB2* transcription in human HEK293 cells [49], suggesting that loss of the hmsLRI-Es, overlapping the microdeletion, can contribute to the pathology by the downregulation of the connected *ZEB2* gene.

### 2.7. Mouse–Human Syntenic Long-Range Interacting Enhancers (hmsLRI-E) Provide Novel Candidate Sequences to Search for Mutations That May Contribute to the Pathogenesis of Neurodevelopmental Disease (NDD)

The degree to which specific NDDs can be explained by mutations within the exome vary greatly for different NDDs. In eye genetic disorders (Microphtalmia/Anophtalmia/ Coloboma-Optic Nerve Hypoplasia, MAC-ONH), only between 40 and 50% of cases are explained by mutations within the exomes of several genes, including *SOX2*, *OTX2* and *PAX6* among the most frequently mutated [50]. Mutations or CNV/rearrangements involving enhancers may thus represent additional causes of disease.

Indeed, in pioneer studies on the genetic basis of eye disease, translocations were found in patients affected by aniridia [4], involving a DNA region which is located, in the normal (wild type) genome, about 150,000 nucleotides 3′ to the *PAX6* gene, whose mutation within the protein-coding exons was already known to cause aniridia. Molecular genetic analysis of aniridia patients bearing CNVs identified a 243.9 kb critical region for *PAX6* transcriptional activation [51], subsequently demonstrated to contain transcriptional enhancers, located within introns of the *Elp4* gene, but in fact controlling expression of *PAX6* [4,52]. One of our hmsLRI-E, (wTR2_9990; chr11:31579239–31580312) corresponding to a previously reported neural enhancer located in an intron of *ELP4* (EnhancerAtlas ID HS042-07410), completely lies within this critical region (Figure 7A); other hmsLRI-E are connected to *Pax6*, located both 3′ and 5′ to *PAX6* (Figure 7B). These sequences may be tested for the presence of mutations (CNV, structural variants or point mutations) in patients affected by NDD, involving the eye.

Our hmsLRI identified about 70 human genomic regions, harboring enhancers (hmsLRI-E), that are connected to eye disease-relevant genes in long-range interaction maps (Appendix A). In particular, the *SOX2* (Figure 8A), *CHD7* (Figure 8B) and *PAX6* (Figure 7B) genes display numerous long-range interactions connecting them to putative enhancers (Appendix A). Further, this set of genes harbors several intronic epigenetic enhancers in mice, that are conserved in humans; in turn, the enhancers are connected to other gene promoters (Appendix A). The LRI perspective proposes these enhancers as candidates for targeted sequencing in those patients with genetic eye defects (MAC-ONH), that did not yet receive a molecular diagnosis based on exome sequencing.

## 3. Discussion

About 98% of our genome does not encode for proteins. Within this non-coding DNA, transcriptional regulatory regions (enhancers) are scattered, often at a great distance from the gene they regulate, and/or within introns of different genes. Indeed, this was the case for around half of the identified enhancer–promoter interactions. Furthermore, identifying the functional connections between enhancers and the genes they regulate will be of great help to understand genome function, in normal development, genetic disease, and evolution. Our study, by re-mapping onto the human genome more than 10,000 enhancer–promoter LRI identified in mouse NSC by RNApolIIChIA-PET, provides a new resource and contribution to these investigations.

### 3.1. The Vast Majority of Mouse Enhancer–Promoter LRI Have Corresponding Syntenic Regions in the Human Genome

In our work, we re-mapped 10,994 “epigenetic” enhancers (marked by H3K4me1/ H3K27Ac enhancer marks), discovered in mouse NSC through their RNApolII-mediated interaction with gene promoters [10], onto the human genome. For most of them (7698), we found a syntenic region in the human genome, encompassing both the enhancer and the promoter connected to that enhancer in mice (Figure 1 and Figure 2, Table 1 and Appendix A), despite the extensive “reshuffling” of genomic DNA sequences observed between mouse and human (Figure 2B,C). Indeed, in Figure 2C we show the links between each genomic mouse region encompassing the original interaction and the syntenic genomic region in the human genome assembly. Importantly, almost all hmsLRI-E carry epigenetic enhancer marks in human neural cells (Table 2), as they do in mice. Moreover, a substantial proportion (over 50%) of hmsLRI are experimentally detected in HiC or PLAC-seq published datasets. As mentioned in the Results section, the failure to attain a higher representation may depend on the variety of cell types and tissues analyzed in those datasets, which would detect only a subset of the total population of interactions existing in the very diverse complex of human neural cells. An additional point is likely related to the technical differences between HiC and RNApolII-ChIA-PET we originally used in our studies in mouse cells. HiC identifies all possible physical interactions within chromatin, giving origin to very large numbers of detected interactions which might enact a great variety of functions, not all of which are necessarily related to gene transcription. In contrast, RNApolII-ChIA-PET selects those interactions that are mediated by active RNApolII, and thus identifies interactions that are related to transcription; these interactions represent a smaller subset than those obtained by HiC. It is possible that, in the presence of a very high number of relatively abundant HiC interactions, many of the RNApolII-ChIA-PET-identified interactions may escape detection. In this context, we analyzed the data reported in four HiC analyses (the same ones as those indicated in Table 2), assessing what percentage of interactions detected in a given dataset are also identified in the sum of the other three datasets. The comparison (see Appendix A) clearly shows that the total overlap of PsychENCODE long-range interactions was less than 53%, indicating that HiC data do not exhaustively represent the full repertoire of long-range interactions in a complex tissue, such as brain matter, comprising hundreds of different cell types. This conclusion is further supported by data in Table 2: if we assume that the cumulative 51% of hmsLRI that find a counterpart in published HiC datasets are genuine, as they are confirmed by independent analyses, it is clear that each of the five HiC datasets detects a very small proportion of the cumulative real long-range interactions, again confirming the conclusions about the incomplete representation of interactions within published HiC datasets discussed above. These observations highlight the importance of combining multiple independent studies with various techniques (HiC, PLAC-seq, ChIA-PET, etc.), and of defined cell types, to approach a complete description of the repertoire of enhancer–promoter interactions in neural cells. On the other hand, for 33 LRI identified in mice, the enhancer and the promoter remapped, in the human genome, onto different chromosomes, and for 493 enhancers, that interact with a promoter in mouse, a syntenic enhancer was remapped in the human genome, but no syntenic promoter was identified (Figure 1, Table 1). We propose that these enhancers, though conserved in terms of sequence, may have changed their usage in evolution, possibly activating different genes in humans and mice. Vice versa, for 2308 gene promoters, connected to enhancers in mice, we fail to detect a syntenic enhancer in humans (Figure 1, Table 1); also in this case, the loss of connected enhancers in humans, or the acquisition of novel enhancers in mice, may be the source of functionally relevant changes in gene activity. Loss or gain of enhancers have been proposed to be an important source of evolutionary change in vertebrates, a hypothesis validated by a few, though meaningful, functional studies [54,55,56], and our synteny maps may help develop novel hypotheses in this field.

### 3.2. Human–Mouse Syntenic Enhancers Involved in Long-Range Interactions (hmsLRI-E) Are Highly Enriched in Candidate Cis-Regulatory Elements Accessible in the Adult Mouse Brain

For the vast majority (about 97%) of the hmsLRI-E identified by our analysis, the mouse syntenic enhancers overlap with candidate cis-regulatory elements (cCRE), identified in mice by their accessible chromatin state in single-nuclei assays performed on a variety of adult mouse brain cell types [11] (Figure 4). This is in a way surprising, since our LRI have been mapped in undifferentiated neonatal NSC. On the other hand, the differentiated cell types, whose cCRE are most enriched within our hmsLRI-E, correspond to the cell types that our NSC generate upon in vitro differentiation: astrocytes, GABAergic neurons, oligodendrocytes (Figure 4). Our findings support the hypothesis that “epigenetic” enhancers, and their LRI with promoters, can mark, in stem cells, sequences that will be later used for gene activation in differentiated neurons and glia [10,12,13,57,58]. Further, the annotation of our hmsLRI-E with the overlapping cCRE (Appendix A), together with the knowledge of the specific cell types in which the cCRE are accessible, will represent a useful resource for future studies aimed at understanding the function of these enhancers in normal development and in disease (see below).

### 3.3. HmsLRI Contain Sequence and Copy Number Variations Associated with Brain Genetic Disease, Providing Potential Insight into Disease Mechanisms

We discovered that non-coding DNA sequence variants, associated by GWAS to brain diseases and traits (schizophrenia; bipolar disorder; intelligence), overlap with some of our hmsLRI-E (Table 3 and Appendix A, Figure 5B). This allows us to hypothesize that the variant itself, or a closely linked mutation, acts by altering enhancer activity onto the connected gene. This is important, because, as already mentioned, LRI connect enhancers to promoters which can be hundreds of thousands, even millions of base pairs away, and can lie in introns of genes different from the gene promoter they actually control (see Introduction). Indeed, our interaction maps allow us to hypothesize the implication, in schizophrenia and bipolar disorder, of the regulation of genes, connected to variant-containing enhancers, that had not been previously involved in this disease (Table 3). For some other hmsLRI-E, the connected gene had already been hypothesized to contribute to the disease (schizophrenia, Table 3) by association studies between disease and polymorphic variants; in two instances, the variants were located directly within the putative candidate gene, whereas in other ones the candidate gene lied within a broad locus implicated by GWAS (Table 3). In addition, a recent study [59] using HiC data alone or in combination with additional criteria revealed some, but not all, of the putative schizophrenia genes identified by our study (Table 3). The fact that polymorphic variation is now detected both in a candidate disease gene indirectly implicated by GWAS, and in a putative enhancer connected by long-range interaction to that gene, strengthens the case for that particular gene being indeed, when mutated or dysregulated, a contributor to the disease. In conclusion, Table 3, also taken in conjunction with Table 2, indicates that previous HiC studies are not exhaustive in identifying candidate schizophrenia genes, and suggests that our work, based on RNApolII-ChIA-PET analysis, provides a useful complement to HiC-based studies (possible reasons for this being discussed already in the previous paragraphs). Having said this, it is appropriate to mention that the functional effect of the targeted perturbation (by deletion or epigenetic repression) of a given enhancer on the expression of the connected gene has been reported only in a few instances, on a small number of enhancers so far (e.g., two out of four connected genes were validated as targets of variant-containing enhancers, by functional enhancer activation/inactivation studies by Rajarajan et al. [59]; one enhancer was validated in Won et al. [15]. After this crucial step, the connected candidate gene will also have to be further validated as a schizophrenia-contributor gene by functional studies of the gene itself in developing neural brain cells.

As for molecular mechanisms, we hypothesize that the sequence variant in the enhancer acts by altering the transcription factor binding to it, in turn leading to deregulation of the connected gene promoter. For enhancers connected to the *TRIM8* gene, we show that some of the variants indeed fall within a transcription factor binding consensus; intriguingly, one variant alters a potential recognition site for *FOXG1*, a TF whose mutation causes an NDD of the Autism disorders spectrum (Figure 5). These data will allow the study, in future experiments, of whether transcription factor binding is indeed altered (by EMSA and chromatin immunoprecipitation assays), and whether this is relevant for the function of the connected gene (by targeted mutagenesis, or epigenetic modulation, of the enhancer via CRISPR-based approaches) [60]. For these variants that do not alter consensus binding sites, it will be possible to perform targeted sequencing of the whole hmsLRI-E in patients carrying the variant, to uncover possible pathogenic mutations in close linkage with the variant itself.

Approximately 97% of our hmsLRI-E overlap with cCRE (Figure 4); it was thus somewhat unexpected to find that, among hmsLRI-E containing variants associated with schizophrenia and bipolar disorders, only 10 out of 32 overlapped with a cCRE (Table 3 and Appendix A). As our hmsLRI-E were identified in neonatal NSC [10], it is possible that the disease-associated variants present in hmsLRI-E, but not in cCRE, affect enhancers active in cells more undifferentiated than those where cCRE are identified (adult brain). Indeed, our hmsLRI-E are enriched in VISTA enhancers, active in the developing mouse nervous system in early and mid-embryogenesis [10] (see also Appendix A). This interpretation is in keeping with the likely early developmental origin of many instances of schizophrenia and bipolar disorders.

On the other hand, some of the GWAS-identified variants, overlapping our hmsLRI-E, do overlap the human counterpart (identified by synteny) of cCRE [11] (Table 3 and Appendix A). We know about the specific cell types of the adult (mouse) brain, where cCRE are in an open chromatin configuration [11]; this will be very useful to formulate hypotheses about the cellular environment where the variant enhancer functions, implying a possible involvement of that cell type in the cellular mechanisms of pathogenesis. In this regard, it is useful to point out that the interactions that we detect in mouse NSC are largely dependent on the *SOX2* transcription factor, as *Sox2* is bound to a vast fraction of the interacting anchors, and *Sox2* deletion leads to the loss of a high proportion of the interactions. This implies that a *Sox2*-dependent transcriptional program might be related to the cell type where our hmsLRI-E are predominantly detected. Indeed, it is interesting to note that the cell types, displaying the cCRE that are most enriched within our mouse enhancers involved in LRI (Figure 4) correspond to cell types that are most affected following *Sox2* conditional deletion in mice (radial glia-like cells; GABAergic neurons and neuroblasts; specific astrocyte types [61,62,63,64,65,66,67].

We further discovered that copy number variants (CNV), associated with neurodevelopmental disease, affect our hmsLRI-E. For example, some microdeletions affect specific hmsLRI-E, though not the connected gene, with a resulting disease phenotype similar to the one resulting from mutations within the gene exons: an example being the deletions affecting an hmsLRI-E connected to *FOXG1* gene (associated to an autism-related syndrome) (Figure 6A). These findings will direct future CRISPR-based experiments of targeted mutations, or epigenetic repression, of the identified enhancers, to test the effect on the expression of the connected gene, and the consequences on cell function (differentiation, survival, etc.). More generally, our maps, and the annotation (Appendix A) of each syntenic human LRI and LRI-E with CNVs in published datasets of CNVs associated with human NDD, will help in the understanding of the specific contribution of enhancer CNVs to the disease.

## 4. Materials and Methods

An overall analysis workflow with details on analysis steps, tools and database resources is reported in Appendix A.

### 4.1. Mapping Mouse LRI-Syntenic Regions within the Human Genome

We used mouse LRIs previously identified by [10] in NSC cultured from mouse neonatal forebrain cultured in vitro. Starting from ChIP-Seq experiments for H3K4me1 and H3K27Ac histone modifications (Appendix A of [10] and ChIA-PET experiments for 3D chromatin interactions (Appendix A of [10], we obtained the list of coordinates of anchors (L, left anchor; R, right anchor) defining mouse LRIs for wTR1, wTR2 and wTR3 replicate experiments (“anchors” are defined as DNA regions associated via long-range interactions in ChIA-PET experiments, see [10] (Figure 1). As a close correspondence had been found between these three replicates, we considered all the interactions previously identified for wTR1, wTR2 and wTR3. From this list, we selected the interactions between promoter (P) anchors (defined as ±2.5 kb of TSS) and non-promoter, genic anchors (G, i.e., overlapping with a gene region) or intergenic anchors (I, i.e., not overlapping with any gene or promoter region). The “TxDb.Mmusculus.UCSC.mm9.knownGeneMouse” Bioconductor package was used for updating the anchors annotation (Appendix A). Non-promoter anchors embedding both H3K27Ac and H3K4me1 (300 bp min length) epigenetic marks with 10 bp min reciprocal overlap were defined “active enhancers” while those carrying only H3K4me1 marked region (300 bp min length) were considered “poised enhancers”. For downstream analysis, we kept only mouse LRIs with enhancer anchors. Both anchor coordinates were separately mapped to human genome by means of LiftOver command line tool [68] using http://hgdownload.soe.ucsc.edu/goldenPath/mm9/liftOver/mm9ToHg19.over.chain.gz (accessed on 15 July 2021) chain file and specifying -minMatch 0.5 which means that more than 50% of bases are lifted over to the human genome).

Subsequently, mouse LRIs for which both anchors were conserved in a syntenic region of the human genome were selected and called “hmsLRI” (Table 1). The identified human syntenic regions were annotated using the UCSC KnowGene track “TxDb.Hsapiens.UCSC. hg19.knownGene” [69] distinguishing among exonic, intergenic, intronic or promoter (Figure 2B). Human syntenic enhancer regions were classified intragenic if they overlapped (1 bp min.) any gene coordinates (5′ UTR, exon, intron, 3′ UTR) while syntenic promoter regions were first checked for overlap with any promoter region (defined as ±2.5 kb of TSS). The conservation level for hmsLRI-E (enhancers involved in hmsLRI) was measured using UCSC phastCons conservation scores for the human genome (hg19) calculated from multiple alignments with 99 other vertebrate species and retrieved in R using Bioconductor package phastCons100way.UCSC.hg19 [70]. A set of random shuffled genomic regions with similar length and chromosomal distribution of hmsLRI were simulated using randomizeRegions() function of regioneR package (Gel, B. et al. (2015) [71].

### 4.2. Comparing hmsLRI-E and hmsLRI with Published Datasets of Epigenetic Enhancer Marks and Long-Range Interactions Previously Determined in Human Neural Cells

For TAD regions we downloaded the DER-18_TAD_adultbrain resource from http://resource.psychencode.org/ (accessed on 4 March 2022) and CP/GZ_TAD.bed files from GEO (GSE77565 dataset). We retrieved histone modification signals (broad inferred peaks) from the Roadmap Epigenomics database https://egg2.wustl.edu/roadmap/web_portal/meta.html, (accessed on 4 March 2022) and checked for an overlap of at least 50 bp with the hmsLRI-E. H3K4me1 peaks corresponding to potential enhancers were available for each neural tissue while H3K27ac only for a subset. Human interactions were from http://resource.psychencode.org/ (accessed on 4 March 2022) (HiC-derived Enhancer-Gene and Promoter linkages, INT-16_HiC_EP_linkages file).

### 4.3. Annotation of hmsLRI and hmsLRI-E by Their Overlap with VISTA Enhancers, cCREs, and Association with NDD

VISTA enhancers: All FASTA sequences of human 998 “positive” and 944 “negative” enhancers with evidence of in vivo activity in developing mouse embryos were downloaded from the VISTA enhancer browser (https://enhancer.lbl.gov/frnt_page_n.shtml (accessed on 29 September 2021), [72,73] and compared with syntenic human enhancer anchors. Human VISTA enhancers had been tested in transgenic mouse embryos [72,73]; each enhancer was defined “positive” if it showed reproducible expression in the same CNS area in at least three independent transgenic embryos, otherwise it was considered “negative”.

Comparison with brain candidate cis-regulatory elements (cCRE): The full list of the 491,818 mouse cCRE along with their genomic coordinates and cell type assignments were retrieved from Appendix A of [11]. We checked for any overlap with our mouse enhancer anchors using findOverlaps function of GenomicRanges.

Gene2Phenotype annotation: Developmental Disorders Genotype-to-Phenotype Database (DDG2P, V.2021.11.12) and EyeG2P (V.2021.10.5) gene lists were downloaded from the Gene2Phenotype (G2P) website (https://www.ebi.ac.uk/gene2phenotype/, accessed on 12 November 2021). In DDG2P we selected only genes associated to eye and brain/cognition (“organ specificity list”), the full list was instead kept for EyeG2P. After matching with hmsLRI associated genes we reported their gene-diseases confidence level of association (Appendix A) as reported by the Gene Curation Coalition (https://thegencc.org/, accessed on 12 November 2022). Gene-disease connection is classified by this database as definitive, strong, moderate, supportive, limited, disputed, refuted, animal or unknown.

### 4.4. Mapping DDD Risk Variants with Respect to hmsLRI-E

We used the subset of GWAS variants with a posterior probability of association (PPA) larger than 1% in schizophrenia [74], bipolar disorder [75] and intelligence [76] previously identified as potential causal variants by [11], which had been overlapped with cCREs. The list of sequence variants was crossed with the hmsLRI-E (Appendix A), and the variants showing overlap are reported in Appendix A, together with the coordinates of the overlapping hmsLRI-E.

The full lists of variant regions of Zarrei’s and Coe’s datasets were downloaded from dbVar study accession number nstd173 and nstd100, respectively. We filtered variant regions with copy number gain/loss (hg19 coordinates) and compared them to hmsLRI using findOverlaps function of GenomicRanges Bioconductor package. AnnotSV vs. 2.2 [77] was used to classify CNV according to their potential clinical impact. Three classes were taken into account: benign (encompassing AnnotSV Class 1 and 2), variant of unknown significance (equivalent to AnnotSV class 3) and pathogenic (encompassing AnnotSV Class 4 and 5) variants.

The study from Coe et al., 2014 [39] is available from dbVar nstd100; it was designed to identify CNVs associated with Developmental delay. In this dataset we have data from individuals with intellectual disability, developmental delay and/or ASD (although the precise phenotype is not specified among the dataset) for a total of 29,085 primarily pediatric cases in comparison to 19,584 adult population controls. The variant calls from Coe et al. [39] were mapped with respect to hmsLRI, to possibly identify rare CNVs that can disrupt the interaction.

## 5. Conclusions

The observations reported in Table 3 and Figure 4 and Figure 5 show that a small, but useful, number of disease-associated sequence variants, detected in hmsLRI-E, allows the identification of connected genes already identified as candidates for a role in brain disease, when mutated, as well as a substantial number of new candidate genes, indicating the usefulness of adding our approach to existing and future HiC-based studies. This implies that, in principle, our strategy may not only improve variant interpretation in known disease-causing genes (by functionally connecting variant-containing enhancers to the correct genes, identified via their interaction with the enhancer), but may also identify novel disease genes which had not been previously identified as disease genes (see cases in Table 3). Overall, long-range interaction maps (and specifically ChIA-PET maps) contribute novel hypotheses allowing us to relate sequence variants to changes in expression of the connected gene (in normal circumstances) to the DNA tract where the mutation (CNV, sequence variant) is located. In view of the ongoing large effort in genome-wide sequencing of patients with neural disease, it is likely that many new mutations in enhancers will be discovered, allowing us to extend our present strategy to the identification of new candidate disease genes. Likewise, a deeper understanding of functional effects of deletions or duplications (CNV) can be obtained by the progressive addition, and analysis, of new cases in which enhancers are involved.

## Figures and Tables

**Figure 1 ijms-23-07964-f001:**
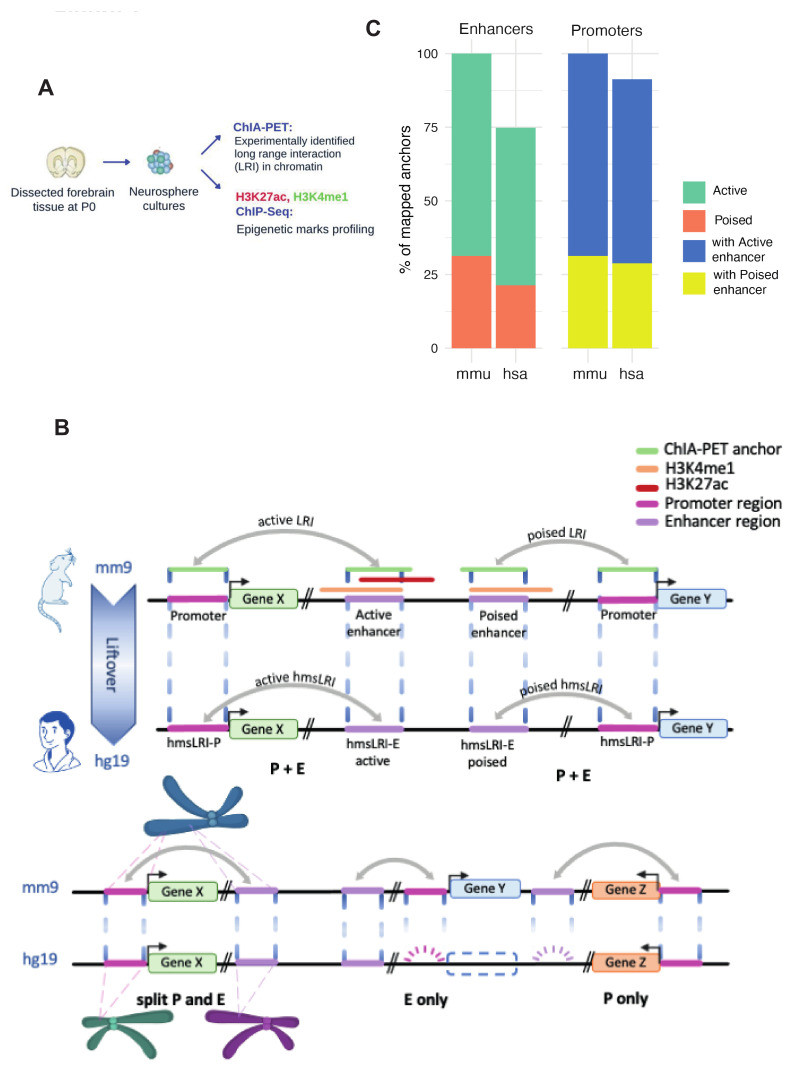
**Identification of human DNA regions syntenic with mouse enhancer–promoter long-range interactions (LRI)**. (**A**) Long-range interactions (LRI) identified by RNApolII ChIA-PET in neural stem cells (NSC) cultured in vitro from mouse neonatal forebrain, and regions carrying H3K4me1 and H3K27Ac histone modifications marks as identified by ChIP-Seq were obtained from (Bertolini et al., 2019) [10] (**B**) Graphical representation of the syntenic mapping between mouse (mm9) and human (hg19) reference genomes. Mouse LRI “anchors” that have syntenic correspondences in the human genome are indicated as hmsLRI-P (promoter) or hmsLRI-E (enhancer). (**C**) The percentages of mouse promoter and enhancer anchors that “lifted over” to the human genome (imposing a 50% minimal overlap, see Methods).

**Figure 2 ijms-23-07964-f002:**
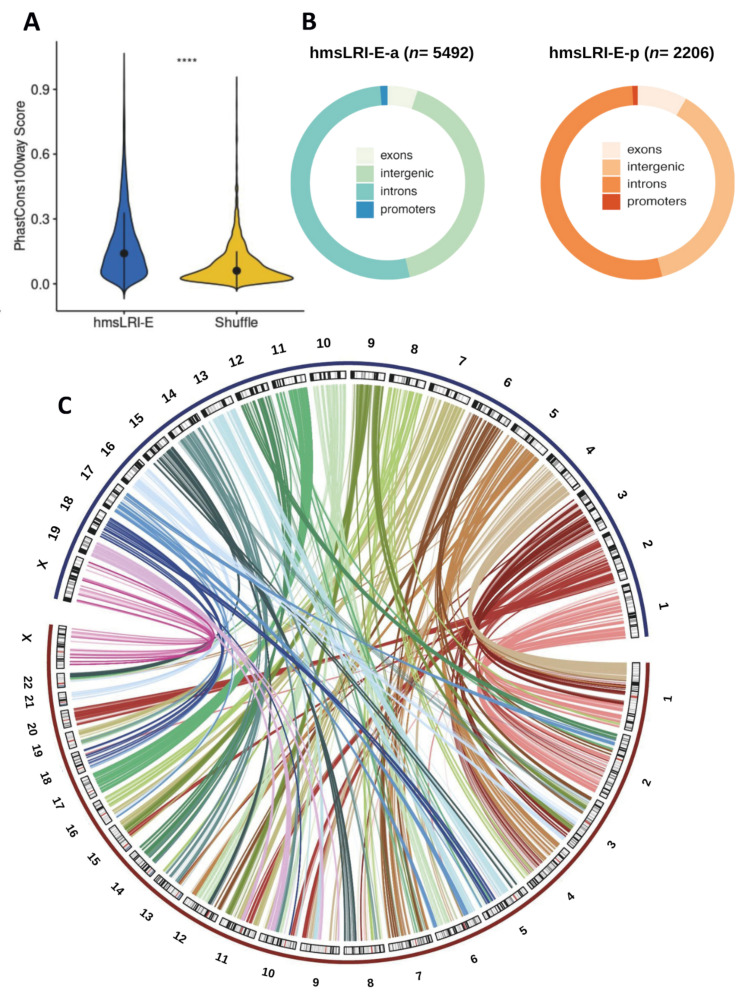
**Conservation and synteny between mouse and human LRI-E and LRI**. (**A**)—Violin plots showing sequence conservation of hmsLRI-E, versus random shuffled genomic regions with similar GC content (**** *p* < 0.0001, Wilcoxon rank-sum test). PhastCons score over 100 vertebrates is reported. Lines span the first to third quartiles (Q1 to Q3), dots denote the median. (**B**) Ring chart with the fraction of hmsLRI-E (active on left and poised on right) overlapping exonic, intergenic, intronic or promoter regions in the human genome. (**C**)—Circos plot showing the genomic correspondences of enhancers involved in LRI between mouse and human. Mouse chromosomes are represented in blue (chr1-19, X), human chromosomes are in red (chr1-22, X).

**Figure 3 ijms-23-07964-f003:**
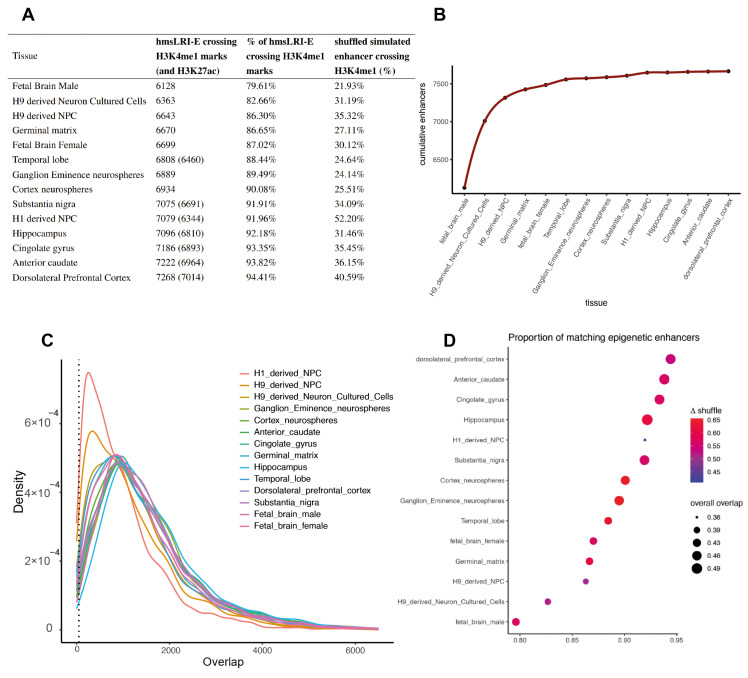
**Concordance between hmsLRI-E and human DNA regions carrying epigenetic enhancer marks (H3K4me1; H3K27ac) from Roadmap Epigenomics**. (**A**)—For each human neural tissue, the number of hmsLRI-E encompassing enhancer marks (H3K4me1; H3K27ac where available) and their relative percentages are shown. The values are compared with the % of shuffled enhancer-like regions overlapping H3K4me1 marks. (**B**)—Cumulative number of matched hmsLRI-E in all the tissues. On the x-axis, tissues are listed from the one with the lowest number of matching hmsLRI-E to the one with the highest number. The y-axis shows the cumulative number of unique hmsLRI-E that progressively match the epigenetic enhancer marks of the indicated tissue (the percentage for each tissue is added to the ones already found for the previous ones). (**C**)—Distribution of the extent of overlap (in base pairs, X-axis) between hmsLRI-E and human regions carrying epigenetic enhancer marks in human cells according to the Roadmap Epigenomic database (“Density” in the Y axis). The dotted line at 50 bp represents the cutoff below which the hmsLRI-E and the enhancer marks were not considered matched. (**D**)—Dot plot representing, on the X-axis, the proportion of hmsLRI-E overlapping human DNA regions (min 50 bp) carrying epigenetic enhancer marks in human neural cells (“matching epigenetic enhancers”), as previously defined in the Roadmap Epigenomic database, in the various tissues indicated on the Y-axis. The width of the dots represents the proportion of bp covered by the methylation region out of the total length of the matched hmsLRI-E. Color is dependent on the difference (delta) between the match of the hmsLRI-E with H3K4me1-marked regions and the match of the simulated shuffled enhancers, in terms of proportion.

**Figure 4 ijms-23-07964-f004:**
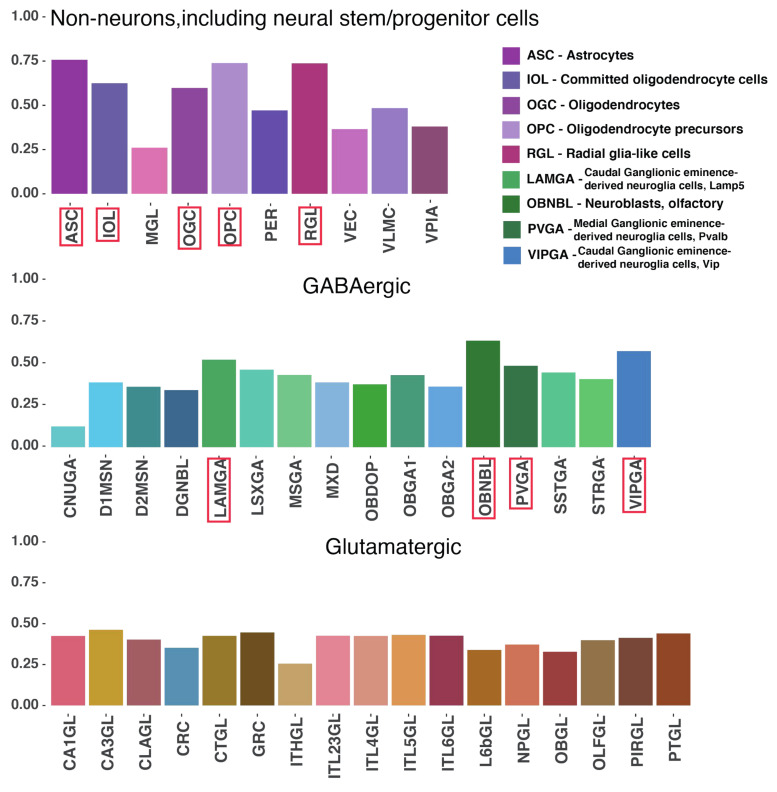
**cCREs are highly represented within mouse enhancers involved in LRIs**. Histograms represent the relative proportion of cCREs (Li et al., 2021) [11] overlapping our mouse enhancers connected to promoters via LRI, for each of the 43 cell type subclasses (each of which is constituted by several different cell types, see Li et al., 2021 [11]. Red rectangles highlight subclasses with more than 50% of cCREs overlapping our mouse enhancers. Top, non-neuronal cell types, including neural stem/progenitor cells, grouped into 10 subclasses. Middle, GABAergic cell types grouped into 16 subclasses. Bottom, glutamatergic cell types grouped into 17 subclasses. For each subclass, mean values for all the different cell types constituting each subclass are reported. The cell type subclasses are defined in Li et al., 2021 [11], Appendix A.

**Figure 6 ijms-23-07964-f006:**
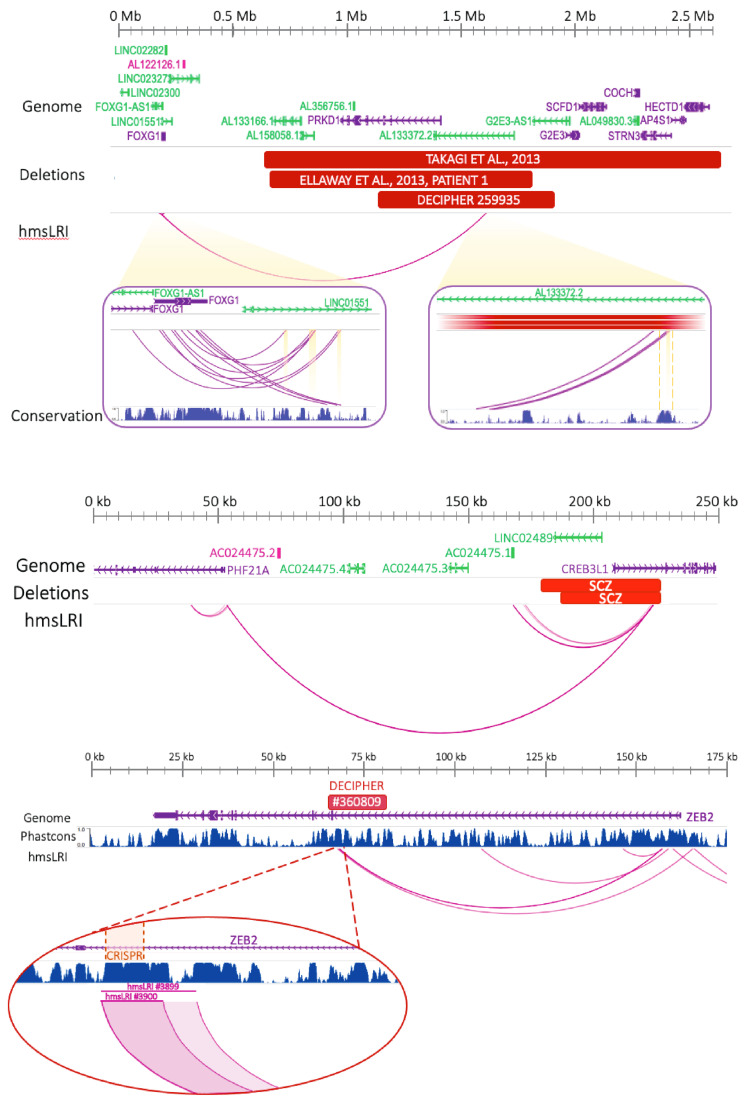
CNV involving hmsLRI-E, but not the interacting gene promoter, suggest that deregulation of the (intact) connected gene may be involved in pathogenesis. (**A**) CNV (microdeletions) identified in NDD patients involve an hmsLRI-E connected to the *FOXG1* promoter. The deletions track represents CNVs harbored by neurodevelopmental delay and Rett-like syndrome patients reported by (Ellaway et al., 2013; Takagi et al., 2013) [36,37] and DECIPHER patient #259935. Magnification of distal and proximal hmsLRI is depicted in the purple boxes; the conservation track shown herein refers to the UCSC genome browser PhastCons score: distal hmsLRI fall into a highly conserved non-coding region. Highlighted hmsLRI overlap with cCREs. (**B**) hmsLRI spanning the *PHF21A*-*CREB31L* locus. Active hmsLRI (wTR2_9924) connecting the promoter of *PHF21A* to an enhancer region located in an intron of *CREB31L*, overlaps two deletions reported in patients affected by schizophrenia (red horizontal rectangles marked SCZ). (**C**) A CNV (14 kb microdeletion, red horizontal box: DECIPHER #360809: chr2:145186591–145201317) affects two hmsLRI-E (wTR1_3899, chr2:145188059–145190658, and wTR1_3900, chr2:145188059–145189733). Phastcons score refers to the UCSC track. (**D**) Magnification of the two hmsLRI. HmsLRI #3900 lies within an extremely conserved region; a notable portion of it (chr2:145188149–145189110), highlighted in orange, has been deleted via *CRISPR* by (Bar Yaacov et al., 2019), proving its role into *ZEB2* transcriptional regulation.

**Figure 7 ijms-23-07964-f007:**
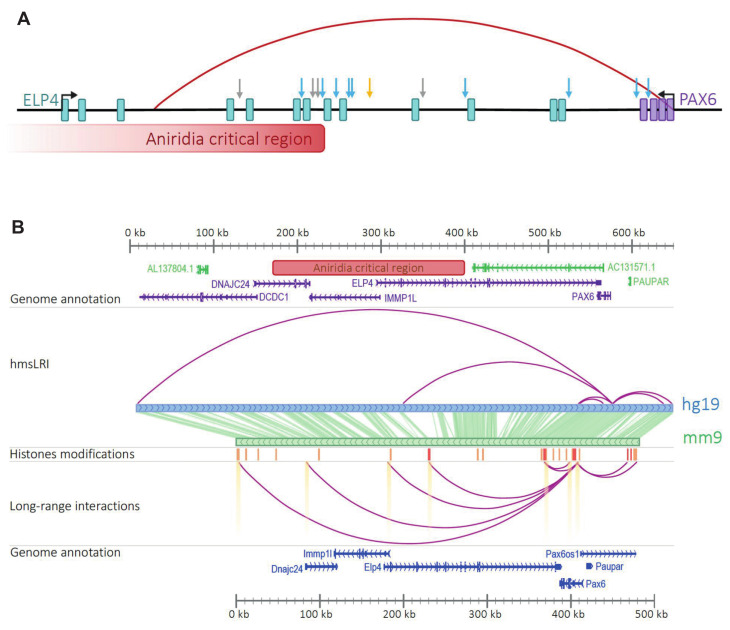
**LRI connects PAX6 to multiple distal enhancers in the aniridia critical region.** (**A**) Blue arrows represent enhancers active in ocular structures, grey ones denote enhancers active in other brain structures; in orange, the SIMO element identified by (Kleinjan et al., 2001; Kleinjan and van Heyningen 2005) [4,53]; the red arch connects PAX6 promoter to an hmsLRI-E. The aniridia critical region, chr11:31422424–31666340, encompasses our enhancer. (**B**) LRI involving the Pax6 promoter reported in (Bertolini et al., 2019) [10]; the track “histone modifications” shows regions carrying marks of poised (orange, H3K4me1) and active enhancers (red, both H3K4me1 and H3K27ac); highlighted LRI overlap with cCREs reported in (Li et al., 2021). In the upper part, long-range interactions, involving active or poised enhancers, present among our hmsLRI-E. Highlighted enhancers and promoters overlap with cCREs reported (Li et al., 2021) [11]. The middle panel is a representation of corresponding regions between the mouse (mm9) and human (hg19) genome.

**Figure 8 ijms-23-07964-f008:**
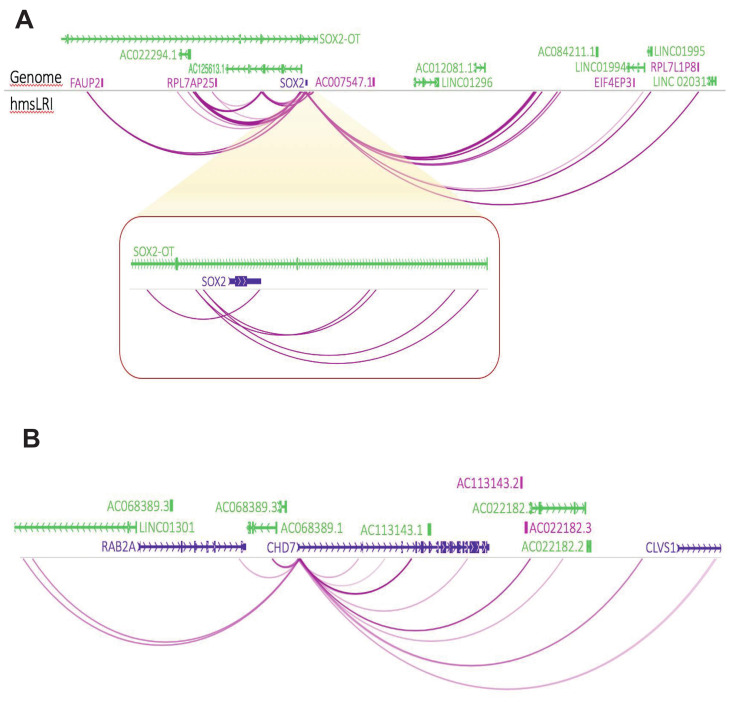
***SOX2*****and*****CHD7*****promoters show intensive connectivity to distal enhancers**. The color intensity of arches is proportional to the number of interactions detected (see PET counts in Appendix A). (**A**) *SOX2* locus. The red box shows a magnification of the more proximal *SOX2* region. (**B**) *CHD7* locus.

**Table 1 ijms-23-07964-t001:** **Human DNA regions identified by their synteny with mouse enhancer–promoter long-range interactions previously identified by RNApolIIChIA-PET**. Total number of mouse LRI connecting promoter and enhancer anchors that were checked for syntenic conservation into the human genome (-minMatch = 0.5 for LiftOver). “Involving Active enhancers” and “Involving Poised enhancers” report interactions in which the connected enhancer carries active and/or poised epigenetic enhancer marks (H3K27Ac, H3K4me1; see Figure 1). Numbers in brackets report the subset of the indicated LRIs which are redundant, i.e., those sharing more than 50% of length for both promoter and enhancer anchors.

Long-Range Interaction (LRI)	Involving Active Enhancer	Involving Poised Enhancer	Total
Mouse Enhancer–Promoter LRI analyzed	7554 (769)	3440 (173)	10,994
Human syntenic LRI identified in human (promoter+enhancer) (“P+E” in Figure 1B)	5492 (501)	2206 (98)	7698
Only enhancer of mouse LRI identified in human (“E only” in Figure 1B)	358	135	493
Only promoter of mouse LRI identified in human (“P only” in Figure 1B)	1355	953	2308
Inter-chromosomal LRI identified in human (“split P and E” in Figure 1B)	26	7	33
Mouse LRI (enhancer, promoter) not represented in human	323	139	462

**Table 2 ijms-23-07964-t002:** **Human putative interactions in this study (hmsLRI) that are found within experimentally determined human interactions datasets in neural cells**. The numbers of hmsLRI represented within the indicated datasets of interactions are indicated, together with the percentage of total hmsLRI (7698, Table 1) represented by these numbers. (*) interactions represented within more than one study were counted only once. CP, cortical plates; GZ, germinal zone; DPC, dorsolateral prefrontal cortex; NPC, neural progenitor cells; PFC, prefrontal cortex; TC, temporal cortex; CB, cerebellum; RG, radial glia; IPC, intermediate progenitor cell; EN, excitatory neurons; IN, interneurons; mgHC, mid-gestational human cortex.

Cells Origin	N of hmsLRI Represented	% of hmsLRI Represented	Study
Hi-C derived CP and GZ of fetal brain	1531	20.67%	[15]
Hi-C interactions for DPC, NPC, Hippocampus	1893	24.59%	[17]
HiC-derived enhancer–promoter linkages of PFC, TC, CB	627	8.14%	[14]
HiC loops associated with pro- moter regions of PFC, TC, CB	788	10.24%	[14]
PLAC-seq derived interactions of RG, IPC, EN, and IN from mgHC	2214	28.76%	[18]
TOTAL (*)	3950	51.31%	

## Data Availability

The LRI maps as WashU Epigenome Browser and UCSC Genome Browser tracks are available at https://github.com/ctglab/hmsLRI_paper, accessed on 18 July 2022 and on Zenodo https://zenodo.org/record/6857114 [78].

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
