# Peer review of "Bridging between Mouse and Human Enhancer-Promoter Long-Range Interactions in Neural Stem Cells, to Understand Enhancer Function in Neurodevelopmental Disease"

_ijms, 2022, doi:10.3390/ijms23147964_

Round 1
Reviewer 1 Report
The authors take advantage of their previous work dedicated to a mouse model of neural stem cell to enlarge their study of long range genomic interactions to human gene regulation and neurodevelopmental diseases. Their bioinformatic methods partially lead to similar results than Hi-C experiments. Their results argue for the usefulness of this syntenic approach to find new mechanisms of diseases at a genomic scale, and seem promising to increase the diagnostic rate in the fields of ID and ASD. The original aspects lie on the syntenic approach using a deep data mining procedure based on GWAS and functional genomics ChIA-PET data. They are able to identify putative active enhancers and long distance interactions with promoters of human genes known to be expressed during neurodevelopment and their potential dysfunction in some neurodevelopmental disorders. This latter part is more challenging.
Minor concerns :
Methods: The authors have to explain the choice of the mouse and human genome versions.
Methods could be summarized for mor eclarity with a global plan or tree of the strategies and corresponding bioinformatic tools and database resources.
Results: Lines 421 to 443. Two CNVs from the Zarrei dataset are discussed. The authors should indicate the structural criteria used to consider the CNVs as potentially relevant, it is doubtful for the 130 kb duplication.
The authors present data on risk variants (in 3.5) as well as data for potential pathogenic CNVs (in 3.6) althouth the underlying pathophysiology and genetics are different. Is their approach more powerful for risk variants ?
Are these interactions only useful to gain insight into disease mechanisms? Could a defined genetic pathology be used as a control to validate their strategy in mutation screening? is it the case for aniridia?
Given the interest of deciphering long distance interactions to discover new genes involved in diseases the authors could also insist on the development of new tools to improve variant interpretation in known genes. These more clinical aspects should be developed before the last sentence in the conclusion.
Author Response
Response to Reviewer 1 Comments
We thank the reviewer for the positive evaluation of our work, and the useful suggestions.
Point 1: Methods: The authors have to explain the choice of the mouse and human genome versions.
Response1: We started our analysis from long-range interactions that had been originally identified in mm9 (Mouse Build 37) and we converted them to human by using UCSC Chain files and liftover tool with the most updated assembly version, in this case it was hg19 (http://hgdownload.soe.ucsc.edu/goldenPath/mm9/liftOver/).
Point2: Methods could be summarized for more clarity with a global plan or tree of the strategies and corresponding bioinformatic tools and database resources.
Response2: We thank the reviewer for the suggestion, and we have added as Supplementary Figure 5 a schematic workflow with details on analysis strategy, tools and database resources.
Point3: Results: Lines 421 to 443. Two CNVs from the Zarrei dataset are discussed. The authors should indicate the structural criteria used to consider the CNVs as potentially relevant, it is doubtful for the 130 kb duplication.
Response3:We hypothesize that deletions affecting an enhancer but not the connected gene promoter might contribute to the pathogenesis (at least in part) by removing an important enhancer contributing to the gene transcriptional output (examples in Fig. 6); conversely, duplications of a DNA region connected to e gene promoter, but not the promoter itself (such as the duplications shown in Fig. S3), might contribute to pathogenesis by increasing enhancer dosage, assuming that the duplicated enhancer carries the sequence information required to interact with that particular gene promoter. A brief explanation has been added in the text (page 16 line 429).
Point4:The authors present data on risk variants (in 3.5) as well as data for potential pathogenic CNVs (in 3.6) althouth the underlying pathophysiology and genetics are different. Is their approach more powerful for risk variants ?
Response4: We propose that our approach can contribute to our understanding of both risk variants and CNV pathogenicity. In both cases, if a DNA variant involves the alteration of an enhancer and/or of its connectivity with the gene it normally regulates, this may contribute to the disease by deregulation of the levels of gene expression.
Point5: Are these interactions only useful to gain insight into disease mechanisms? Could a defined genetic pathology be used as a control to validate their strategy in mutation screening? is it the case for aniridia?
Response5: Indeed, as the reviewer noted, we detected enhancers connected to the aniridia-causing gene PAX6 within a DNA region that had already been already described to contain enhancers, whose mutation can cause aniridia; in this sense, this is a positive control (see Fig. 7). Also in the case of schizophrenia, the fact that we detect enhancer anchors, that overlap with sequence variants, previously associated with schizophrenia, and that are connected to genes previously reported to contribute to schizophrenia when mutated, constitute a validation of our approach. This is discussed in the text in Results and Discussion, regarding the data in Table 3.
Point6:Given the interest of deciphering long distance interactions to discover new genes involved in diseases the authors could also insist on the development of new tools to improve variant interpretation in known genes. These more clinical aspects should be developed before the last sentence in the conclusion.
Response6:We thank the reviewer for the suggestion. We added a phrase in the Conclusion (page 23, line 662, 666).
Reviewer 2 Report
This research article has investigated the enhancer-promoter long-range interactions in the human genome compared with that in the mouse genome. The results provide some important insights into understanding how enhancers can regulate human DNA regions by connecting to identified 7 promoters via long-range interactions. The authors worked out the profiles in different cell types. The study is well organised, and the obtained results can be helpful for further research progress in the regulation of human DNA regions, especially in neuropathological conditions. I support publishing this paper, which requires some grammar editing of the text and improving the style. Also, some figures require a re-assessment of their clarity, such as Figure 2C. Adding a list of abbreviations will be useful as the text contains numerous abbreviations, making it difficult to read.
Author Response
We thank the reviewer for the positive evaluation.
Response1:Figure 2C shows the “reshufflling” of genomic DNA sequences encompassing LRI between mouse and human assemblies. We added the reference to it in the text (page 19, line 514), and the requested list of abbreviations (page 23-24, line 694)
We re-read the manuscript regarding grammar and style.